# Study and Application of Rock Drilling Resistance Characteristics in the Jiyang Depression Formation

Xiaoyong Ma [1], Wei Cheng [2,3,*] and Liang Zhu [2,3]

1 Yellow River Drilling Company, Sinopec Shengli Petroleum Engineering Corporation, Dongying 257000, China; maxy999rz@163.com
2 School of Petroleum Engineering, Yangtze University, Wuhan 430100, China; zhuliang123@yangtzeu.edu.cn
3 Hubei Provincial Key Laboratory of Oil and Gas Drilling and Production Engineering, Wuhan 430100, China
* Correspondence: cw05241989@163.com

**Abstract:** In response to the unclear drilling resistance characteristics of rocks in the Ji'yang Depression, low drilling efficiency of PDC drill bits, and difficulties in drill bit selection, this study selected rock samples from different depths in the area for indoor drilling resistance analysis testing. Based on logging data, a prediction model was established for drilling resistance characteristics parameters of the strata in the area, and a graph of drilling resistance characteristic parameters of the rocks in the area was drawn. The study showed that the uniaxial compressive strength of the strata rocks was 50–110 MPa, with a hardness of 500–1300 MPa, a plasticity coefficient ranging from 1 to 2, a rock drillability grade of 8–20, and an abrasiveness index of 5–20. Combining the analysis of on-site drilling bit failures, PDC drill bits adapted to the strata in the area were selected, and the mechanical drilling speed of the selected bits reached 12.58 m/h, successfully drilling through the target layer. The above research results are of guiding significance for understanding the reasons for the difficulty of drilling into the Jiyang Depression strata and for improving mechanical drilling speed and drill bit selection in this area.

**Keywords:** drilling resistance characteristics parameters; logging data; failure analysis; drill bit selection





## 1. Introduction

The Jiyang Depression is a continental basin that developed against the background of block faulting and the disintegration of the Paleozoic and Precambrian during the Mesozoic Era. Drilling results indicate that the lithology of this area is mainly composed of sedimentary rocks such as sandstone and mudstone, with conglomerate in the Middle and Upper Paleozoic and carbonate rocks in the Lower Paleozoic forming a structurally complex combination, characterized by deep burial of oil layers [1]. The drill bit is an important tool for rock breaking and well completion during the drilling process. Conventional PDC drill bits are mainly composed of a bit body and PDC cutting teeth. The bit body is primarily divided into two categories: cemented carbide matrix and steel body. Meanwhile, PDC cutting teeth are mainly composed of polycrystalline diamond compact (PDC) cutters and cemented carbide, which are sintered at high temperatures and pressure. PDC cutting teeth have excellent wear resistance but poor impact resistance, requiring good compatibility with the formation to improve drilling efficiency during use. Selecting high-quality drill bits is crucial for reducing drilling costs and improving drilling efficiency [2].

Based on the analysis of existing drilling data, the main constraints on accelerating drilling efficiency in this area are manifested by long drilling cycles, relatively low mechanical drilling speeds, and high drilling costs. Therefore, addressing the challenges encountered in drilling engineering in this area, the urgent need is to figure out how to further increase single-well production and reduce project costs, with further optimization of drilling technology schemes being particularly important. The key to accelerating drilling efficiency largely depends on whether the drilling tools, mainly including drill bits

and drilling equipment, are appropriate. Therefore, in conducting related research, the key core issue is the lack of a clear understanding of the distribution pattern of rock drilling resistance characteristics parameters in the Jiyang Depression area, leading to a lack of specificity in drill bit selection. Research has revealed that some layers in this block exhibit low mechanical drilling speeds during drilling, with significant drill bit wear requiring frequent changes, severely affecting drilling cycles and drilling quality. Therefore, it is necessary to clarify the physical and mechanical properties of the corresponding layers in this block based on limited core and logging data.

The rock drilling resistance is a comprehensive indicator for evaluating rock fragmentation by drill bits, which is of great significance for the study of rock characteristics and the selection of drill bit types. It is mainly characterized by parameters such as rock compressive strength, drillability, hardness, plasticity coefficient, and abrasiveness [3]. Many scholars at home and abroad have studied the parameters of rock drilling resistance for a long time. Xiaofeng Xu determined the mineral composition, hardness, plasticity, and abrasiveness of rocks through laboratory tests; the results showed that the integrity, density, and high hardness of rocks were the main factors leading to the low rock-breaking degree of PDC drill bits [4]. Hui Zhang conducted drillability and acoustic property experiments in different drilling directions using a drillability testing device and a rock acoustic velocity measurement device, and established a drillability prediction model for shale formations in different drilling directions [5]. Yang Guan studied the rock mechanics characteristics of mudstone through experimental methods, and the results showed that with the increase of confining pressure, the strength of the rock increased, and the critical transformation pressure of mudstone was obtained [6]. Liu Bin conducted experiments on the drillability, abrasiveness, and drilling resistance mechanism of rocks by selecting rock samples from the lower part of the Xujiahe Formation (Xu'er and Xu'yi sections) in the block and outcrop, and the results showed that the tested rock samples exhibited extremely strong drilling resistance [7]. Gao Li studied the relationship between rock drillability and compressive strength, Poisson's ratio, and elastic modulus using experimental methods. The results indicate that, under confining pressure, there is a good correlation between rock drillability and compressive strength. Under uniaxial compression conditions, when the compressive strength varies between 30 MPa and 60 MPa, rock drillability shows a good linear correlation with Young's modulus or Poisson's ratio [8].

Although many experts and scholars have conducted studies on the rock drilling resistance characteristics through laboratory experiments and logging interpretation methods, obtaining some commonalities among different types of rocks, the physical and mechanical properties of rocks vary at different depths in each block, and the presence of bedding and fractures in certain blocks leads to significant differences in properties. Therefore, the most commonly used approach is still the combination of laboratory physical experiments and logging data, allowing us to obtain relatively accurate parameters of rock drilling resistance characteristics specific to particular blocks. This study conducted laboratory tests on cores taken from the Jiyang Depression, combined with logging data, to establish profiles of rock drilling resistance characteristics, aiming to provide a reliable basis for drill bit selection and drilling speed improvement in this area.

## 2. Formation Drillability Test

### 2.1. Uniaxial Compressive Strength Test

Under the action of uniaxial compression load, the maximum stress that a rock can withstand when it reaches failure is called the uniaxial compressive strength of the rock [9]. Laboratory experiments on the rock's resistance to drilling characteristics were conducted in accordance with the standard DZ/T 0276-2015 [10] Experimental Regulations for Rock Physical Mechanics Properties to obtain parameters of the rock's resistance to drilling. Firstly, rock core sampling was performed using an online cutting machine, with the specifications of the rock core being $\Phi25$ mm $\times$ 50 mm. The experimental equipment used the TAW-2000 rock mechanics testing system, The equipment is produced by Changchun Chaoyang Testing Ma-

chine Co., Ltd. in Changchun, China and is capable of conducting rock mechanical parameter tests under conditions of up to 2000 kN axial force, which can conduct uniaxial compressive strength(Pucs) tests on rock samples, as shown in Figure 1. During the experiment, a layer of lubricant (Vaseline) was applied to the upper and lower surfaces of the specimen, the specimen was placed at the center position of the press head, pressure was applied through the press head on the device, and it was loaded at a rate of 0.2 MPa per second until the specimen failed. The calibrated sensors on the equipment decouple the electrical signals, which are then transmitted to the computer. Through data analysis software "text.23" on the computer, the rock's compressive strength can be read directly.

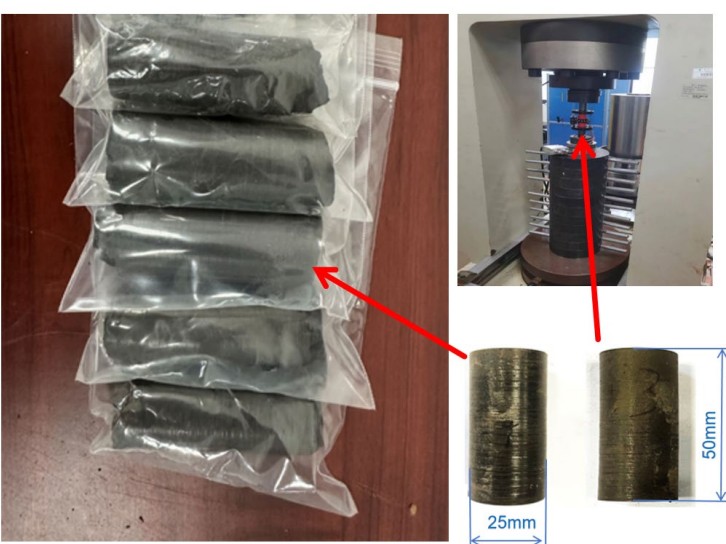

**Figure 1.** Uniaxial compressive strength test.

*2.2. Hardness and Plasticity Tests*

　　The process of rock fragmentation at the bottom of a well during drilling is extremely complex. The crushing load is not static but dynamic, and both the magnitude and direction of the crushing load change over time. Therefore, efforts have been made to simulate the actual conditions at the bottom of the well and study the rock fragmentation process and influencing factors in the laboratory. The hardness and plasticity coefficient of the rock are determined using the cylindrical indentation method. Due to the representative nature of rock fragmentation during the indentation process, the mechanical properties of the rock measured by the indentation method can, to some extent, reflect the rock's resistance to fragmentation during drilling.

　　Rock hardness refers to the ability of a rock to resist penetration or intrusion by other objects on its surface [11]. Hardness experiments were conducted on rock samples using a hardness parameter tester. Under a certain pressure, the same rock sample was subjected to pressure experiments with the same pressure head. By recording the pressure applied to the rock when it was destroyed and the area of the pressure head, the hardness of the rock can be determined. The numerical value can be directly read from the rock hardness tester, as shown in Figure 2. The classification of rock hardness is shown in Table 1; referring to the classification table of rock hardness can determine the hardness level of the rock.

$$H_y = \frac{P}{A} \tag{1}$$

where $H_y$ represents the hardness of the rock, measured in MPa; $P$ denotes the load applied by the pressure head at brittle failure, measured in Newtons; and $A$ signifies the bottom area of the pressure head, measured in square millimeters (mm$^2$).

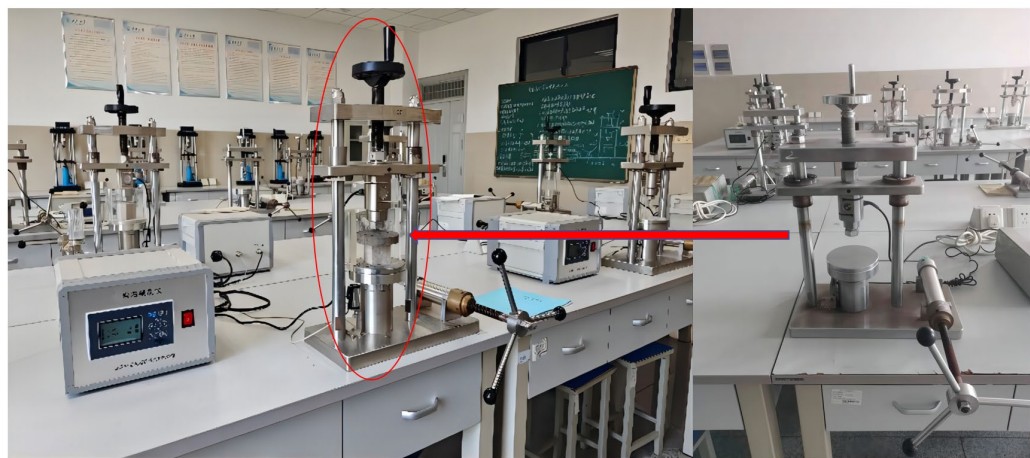

**Figure 2.** Rock hardness test.

**Table 1.** Classification of rock hardness [12].

| Hardness, ×100 MPa | Grade | Category |
|:---:|:---:|:---:|
| ≤1 | 1 | Soft |
| 1~2.5 | 2 | |
| 2.5~5 | 3 | Medium-soft |
| 5~10 | 4 | |
| 10~15 | 5 | Medium-hard |
| 15~20 | 6 | |
| 20~30 | 7 | Hard |
| 30~40 | 8 | |
| 40~50 | 9 | Very hard |
| 50~60 | 10 | |
| 60~70 | 11 | Extremely hard |
| ≥70 | 12 | |

The process of rock deformation under external forces until failure can be divided into three types, as shown in Figure 3. Figure 3a represents brittle rock, characterized by the OD segment as the elastic deformation stage, followed by brittle failure after reaching point D. Figure 3b represents ductile-brittle rock, where the OA segment is elastic deformation, the AB segment is plastic deformation, and brittle failure occurs at point B. Figure 3c represents plastic rock, where plastic deformation occurs under a small load, and its deformation increases with time without any obvious plastic failure phenomenon.

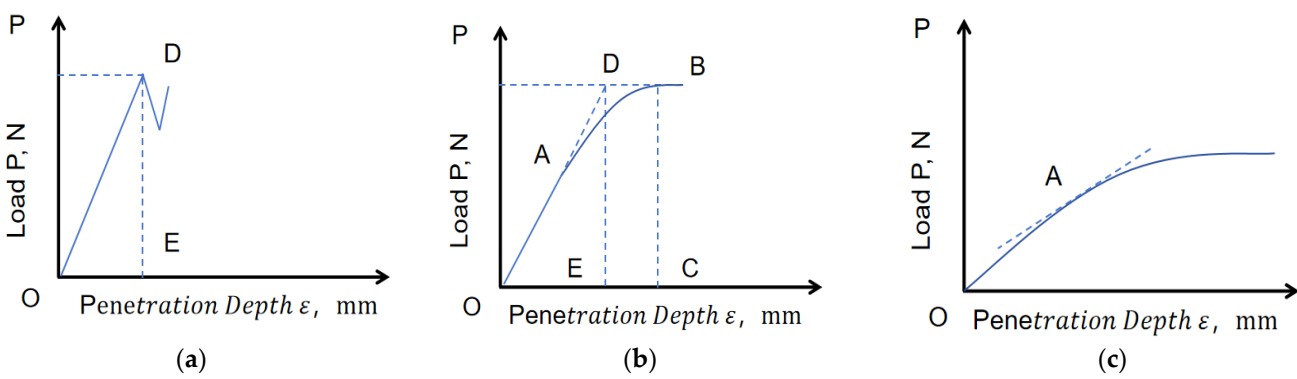

**Figure 3.** Three different types of rock fragmentation processes under external forces: (**a**) Brittle rock; (**b**) Ductile–brittle rock; (**c**) Ductile rock.

The plasticity coefficient refers to the ratio of the total energy consumed during the initial volume fracture of the rock to the elastic deformation energy after the rock is subjected to the pressure head indentation.

$$K_p = \frac{A_F}{A_E}$$ (2)

Here, $A_E$ represents the total energy before rock fracture and $A_F$ represents the elastic deformation energy consumed before rock fracture.

The quantitative grading criteria for rock plasticity coefficients are shown in Table 2. Similar to the method used for rock hardness testing described above, the test for rock plasticity coefficients utilizes the cylinder indentation method. In this method, a flat-bottomed cylindrical head with a certain diameter is used to indent the surface of the rock chips. As the load increases, the depth of the head's penetration into the rock chips gradually increases until the rock chips undergo the initial volume fracture. The plasticity coefficient of the rock is obtained by measuring the load-displacement curve.

**Table 2.** Classification table of rock plasticity coefficients [12].

| Category | Brittle | Brittle-Ductile Transition | | | | Ductile |
|---|---|---|---|---|---|---|
| | | Low Plasticity–High Plasticity | | | | |
| Level | 1 | 2 | 3 | 4 | 5 | 6 |
| Plasticity Coefficient $K_p$ | 1 | >1~2 | 2~3 | 3~4 | 4~6 | >6~∞ |

*2.3. Abrasiveness Test*

Based on the drilling records of the Jiyang Depression, it was found that PDC (Polycrystalline Diamond Compact) drill bits were exclusively used for all layers. Therefore, this experiment employed PDC micro-drilling to test the drillability of the rock. Micro-drilling is a common method used for determining rock drillability in laboratory settings [13]. The experiment needs to meet the requirements outlined in the industry standard "Methods for Determining and Grading Rock Drillability" (S/Y/T 5426-2016 for the test) [14]. Simulated micro-drill bits are used to conduct drilling tests on rock cores according to certain procedures, as is shown in Figure 4. The pre-drilled depth is 1 mm, with the target depth being 3 mm. The drilling pressure is set at 500 N and the rotation speed is 55 r/m. This method can determine the drillability grade values of PDC drill bits and roller cone drill bits under laboratory conditions. The table of PDC bit rock drilling ability classification is shown in Table 3. The main measurement involves recording the time taken for the same type of micro-drill bit to drill to the same depth under certain drilling pressure and rotation speed conditions. The calculation formula is as follows:

$$K_d = log_2 t_d + G_i$$ (3)

where $k_d$ is the drillability grade value; $t_d$ is drilling time in seconds; $G_i$ is the equivalent conversion grade value; and $i$ is the drilling pressure level (first level is 500 KN; second level is 1000 KN; third level is 1500 KN).

**Table 3.** Classification of rock drillability for PDC drill bits [15].

| Number | Grade | Kd | Drilling Time t, s | | | Classification |
|---|---|---|---|---|---|---|
| | | | First | Second | Third | |
| 1 | I | Kd < 2 | t < 22 | | | I (Soft) |
| 2 | II | 2 ≤ Kd < 3 | 22 ≤ t < 23 | | | |
| 3 | III | 3 ≤ Kd < 4 | 23 ≤ t < 24 | 22 ≤ t < 23 | | |
| 4 | IV | 4 ≤ Kd < 5 | 24 ≤ t < 25 | 23 ≤ t < 24 | | |

**Table 3.** *Cont.*

| Number | Grade | Kd | Drilling Time t, s | | | Classification |
| --- | --- | --- | --- | --- | --- | --- |
| | | | First | Second | Third | |
| 5 | V | 5 ≤ Kd < 6 | 25 ≤ t < 26 | 24 ≤ t < 25 | 22 ≤ t < 23 | |
| 6 | VI | 6 ≤ Kd < 7 | 26 ≤ t < 27 | 25 ≤ t < 26 | 23 ≤ t < 24 | II (Medium) |
| 7 | VII | 7 ≤ Kd < 8 | | 26 ≤ t < 27 | 24 ≤ t < 25 | |
| 8 | VIII | 8 ≤ Kd < 9 | | | 25 ≤ t < 26 | |
| 9 | IX | 9 ≤ Kd < 10 | | | 26 ≤ t < 27 | III (Hard) |
| 10 | X | Kd ≥ 10 | | | | |

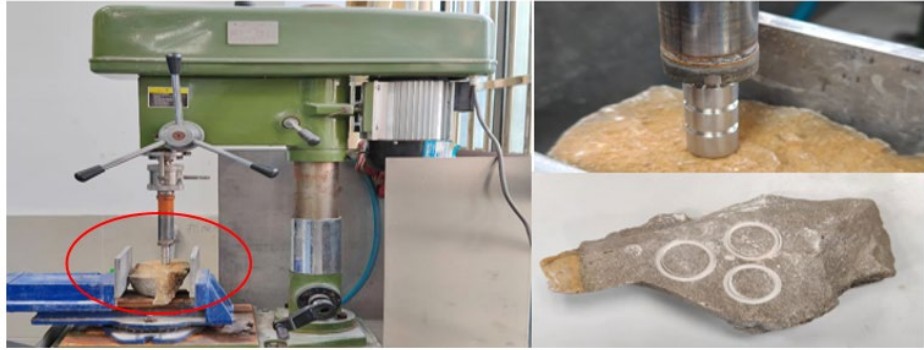

**Figure 4.** Rock abrasion test.

## 3. Geological Drillability Analysis

### 3.1. Modeling of Geological Drillability Parameters

The tests were conducted on 20 rock cores taken from the Jiyang Sag block, and some of the results combined with logging data are shown in Table 4, where Pucs represents compressive strength, MPa; Hy represents hardness, MPa; Kp denotes the plasticity coefficient, dimensionless; Kd indicates the drillability index, dimensionless; GWL stands for abrasiveness index, g/m; and AC represents acoustic time difference, μs/m.

**Table 4.** Correspondence of formation rock drilling resistance parameters with logging data.

| Number | Depth/m | Drilling Resistance Characteristics Parameters | | | | | Logging Data |
| --- | --- | --- | --- | --- | --- | --- | --- |
| | | Pucs (MPa) | Hy (MPa) | Kp | Kd | GWL (g/m) | AC (μs/m) |
| 1 | 3048.32 | 46.28 | 116.57 | 1.02 | 2.55 | 17.6 | 279.3495558 |
| 2 | 3052.43 | 48.90 | 262.78 | 1.10 | 2.81 | 24.3 | 256.3234251 |
| 3 | 3056.6 | 42.60 | 280.43 | 1.04 | 3.41 | 26.3 | 252.512415 |
| 4 | 3062.45 | 50.13 | 315.64 | 1.21 | 3.85 | 25.4 | 255.7313221 |
| 5 | 3113.9 | 49.60 | 470.28 | 1.13 | 4.40 | 24.8 | 250.0777333 |
| 6 | 3120.67 | 47.28 | 133.27 | 1.23 | 4.13 | 22.7 | 266.1806543 |
| 7 | 3140.69 | 53.80 | 156.32 | 1.03 | 3.29 | 19.5 | 275.7690876 |
| 8 | 3157.19 | 54.40 | 425.1217 | 1.12 | 4.84 | 25.4 | 252.7071961 |
| 9 | 3175.33 | 49.20 | 383.842 | 1.15 | 4.46 | 24.6 | 246.5077964 |
| 10 | 3186.47 | 52.10 | 640.39 | 1.13 | 5.46 | 28.6 | 245.4569892 |
| 11 | 3222.10 | 55.96 | 713.24 | 1.22 | 4.83 | 33.5 | 237.4300000 |
| 12 | 3256.54 | 57.70 | 752.61 | 1.26 | 5.39 | 35.5 | 234.3725875 |
| 13 | 3280.32 | 56.32 | 925.36 | 1.14 | 5.75 | 32.6 | 229.6540643 |
| 14 | 3300.67 | 62.10 | 764.32 | 1.15 | 5.30 | 31.5 | 243.5846692 |
| 15 | 3314.55 | 57.28 | 935.93 | 1.26 | 6.44 | 30.6 | 237.2523485 |
| 16 | 3350.46 | 60.78 | 1148.75 | 1.25 | 6.24 | 28.8 | 230.6710892 |
| 17 | 3380.70 | 67.28 | 1263.12 | 1.03 | 6.95 | 29.9 | 225.2970828 |
| 18 | 3392.60 | 63.96 | 1460.58 | 1.07 | 7.67 | 32.3 | 227.8445719 |
| 19 | 3410.78 | 66.32 | 1572.16 | 1.28 | 8.38 | 34.6 | 224.4036872 |
| 20 | 3460.30 | 70.50 | 1686.33 | 1.15 | 8.60 | 36.5 | 225.7457129 |

Rock physics experiments can accurately describe the properties and characteristics of core samples from geological formations. However, it is not feasible to obtain the properties of entire wellbore sections through core testing; only single-point data can be obtained through these experiments [16]. Logging data, on the other hand, possess continuous characteristics. By establishing the corresponding relationship between logging data and rock mechanics parameters, continuous profiles of rock mechanics parameters for geological formations can be calculated [17].

In order to establish the relationship between rock mechanics and acoustic transit time, firstly—according to the theory of elastic waves—dynamic elastic modulus and dynamic Poisson's ratio values are obtained using longitudinal and shear wave acoustic transit time logging data and density logging data. The calculation model is shown in Formulas (4) and (5) [18].

$$E_d = \frac{\rho}{\Delta t_s^2} \frac{\left(3\Delta t_s^2 - 4\Delta t_p^2\right)}{\left(\Delta t_s^2 - \Delta t_p^2\right)} \tag{4}$$

$$\mu_d = \frac{\Delta t_s^2 - 2\Delta t_p^2}{2\left(\Delta t_s^2 - \Delta t_p^2\right)} \tag{5}$$

where $E_d$ represents the elastic modulus, measured in MPa; $\mu_d$ represents the Poisson's ratio, which is dimensionless; $\rho$ represents the formation density, measured in g/cm$^3$; $\Delta t_p$ represents shear wave travel time; and $\Delta t_s$ represents compressional wave travel time, measured in μs/m.

Since there are no direct data on clay content in the logging data, only gamma-ray logging data are available. Therefore, clay content can be calculated using empirical formulas. The formulas for calculating clay content are shown in Formulas (6) and (7) [19].

$$\text{Clay content}: \ V_{cl} = \frac{2^{GCUR.I_{GR}} - 1}{2^{GCUR.I_{GR}}} \tag{6}$$

$$\text{Clay content index}: \ I_{GR} = \frac{GR - GR_{min}}{GR_{max} - GR_{min}} \tag{7}$$

where *GCUR* is the Hillquist index, which is related to geological ages and can be statistically determined based on core data and natural gamma-ray logging values. For the Tertiary strata, the value is 3.7, and for older strata, the value is 2; here, we take 2.0.

According to empirical formulas proposed by domestic researchers for calculating rock uniaxial compressive strength, the uniaxial compressive strength of the target reservoir oil shale can be obtained by regression analysis using the results of uniaxial compressive strength tests and corresponding depth logging data. The calculation model is shown in Formula (8).

$$\text{Pucs} = 0.0492(1 - 2\mu_d)\left\{\frac{1 + \mu_d}{1 - \mu_d}\right\}^2 \rho^2 V_P{}^4(1 + 0.78V_{cl}) \tag{8}$$

In the equation, Pucs represents the uniaxial compressive strength of the rock, measured in MPa; $\mu_d$ represents the dynamic Poisson's ratio, dimensionless; $V_{cl}$ represents the clay content, dimensionless; $V_p$ represents the longitudinal wave velocity, measured in km/s.

After analyzing the experimental data and acoustic logging information, it can be observed that there is a good correlation between the acoustic transit time and rock hardness, plasticity coefficient, drillability rating, and abrasiveness index. The graphical relationships between acoustic transit time and hardness, plasticity coefficient, drillability rating, and abrasiveness index are shown in Figures 5–8. After curve fitting, the fitting equations for rock hardness, plasticity coefficient, drillability grade, and abrasiveness index are obtained as shown in Formulas (9)–(12).

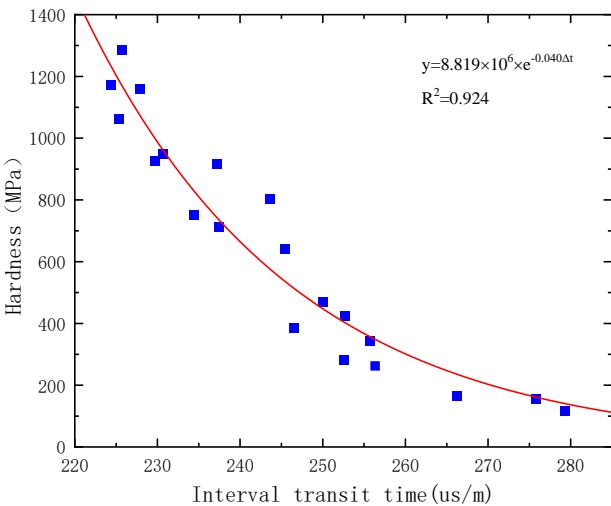

**Figure 5.** Relationship between acoustic time difference and rock hardness.

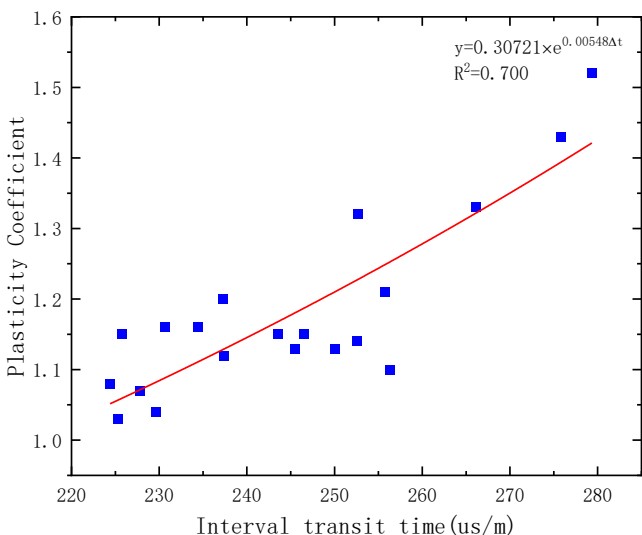

**Figure 6.** Relationship between acoustic wave time difference and rock plasticity.

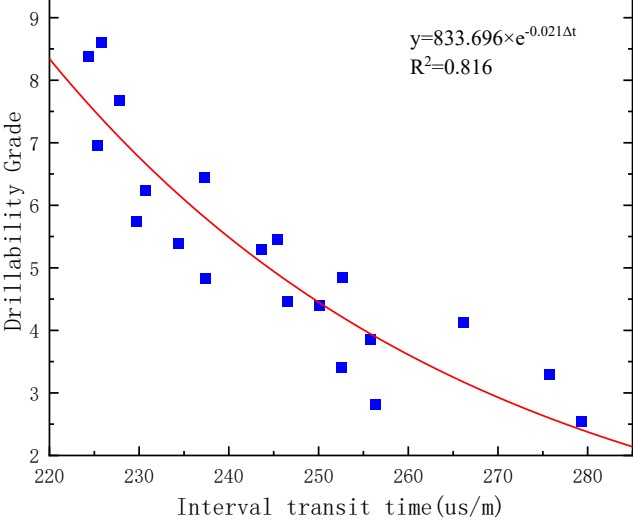

**Figure 7.** Relationship between acoustic time difference and rock drillability index.

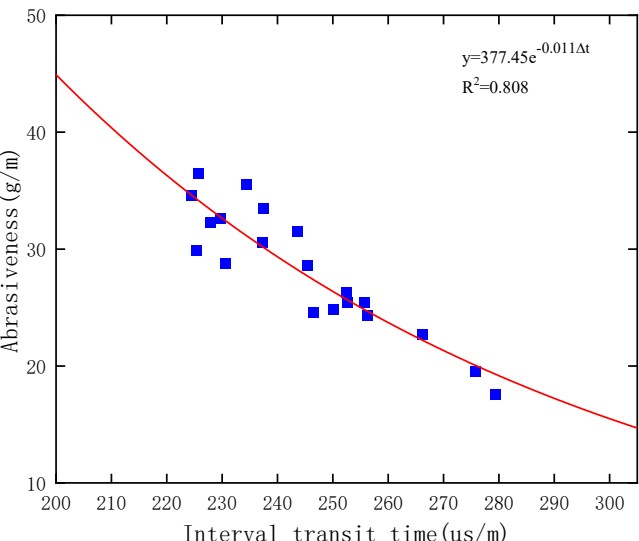

**Figure 8.** Relationship between acoustic time difference and rock abrasiveness index (GWL).

The rock hardness calculation model is represented by Equation (9)

$$H_y = 8.819 \times 10^6 \times e^{-0.040\Delta t} \tag{9}$$

The rock plasticity coefficient calculation model is represented by Equation (10)

$$K_p = 0.30721 \times e^{0.00548\Delta t} \tag{10}$$

The rock drillability index calculation model is represented by Equation (11)

$$k_d = 833.696 \times e^{-0.021\Delta t} \tag{11}$$

The rock abrasiveness index is represented by Equation (12)

$$GWL = 377.45 \times e^{-0.011\Delta t} \tag{12}$$

### 3.2. Profile of Rock Drillability Parameters

Using the calculated models based on the rock drilling resistance characteristics parameters established from laboratory experiments, combined with field logging data, profiles of the compressive strength, hardness, plasticity coefficient, drillability rating, and abrasiveness index of rocks in the Jiyang Depression formation were established, as shown in Figure 9.

From the profiles of rock drilling resistance characteristics parameters, the following conclusions can be drawn:

(1)  In the depth range of 3000–3500 m, there is a significant fluctuation in the rock drilling resistance characteristics parameters, indicating strong anisotropy of the rocks in this area, with greater variation in the lower layers compared to the upper layers. The abrupt change in drilling resistance parameters within this range may be attributed to variations in lithology or changes in horizontal stresses induced by bedding planes.

(2)  Within the range of 3000–3500 m, the uniaxial compressive strength of rocks in this interval ranges from 40 to 65 MPa, the hardness ranges from 120 to 1400 MPa, the plasticity coefficient ranges from 1 to 2, the PDC drillability rating ranges from 3 to 8, and the abrasiveness index ranges from 15 to 36 g/m. According to the classification table, the rocks in this interval are classified as a soft to medium-hard level, with a medium to medium-high level of abrasiveness and, overall, exhibit low plasticity based on the plasticity coefficient.

(3) Overall, with the change in well depth, the patterns of change in the compressive strength, rock hardness, drillability rating, and abrasiveness index of rocks are consistent, while the pattern of change in the plasticity coefficient is the opposite. As the well depth increases, the difficulty of drilling increases, and field drilling records show that the mechanical drilling speed in this interval is relatively low, especially near the bottom of the well at around 3350 m, where the mechanical drilling speed is as low as 1.13 m/h. This is due to the higher drillability rating and stronger abrasiveness of the rocks in this location.

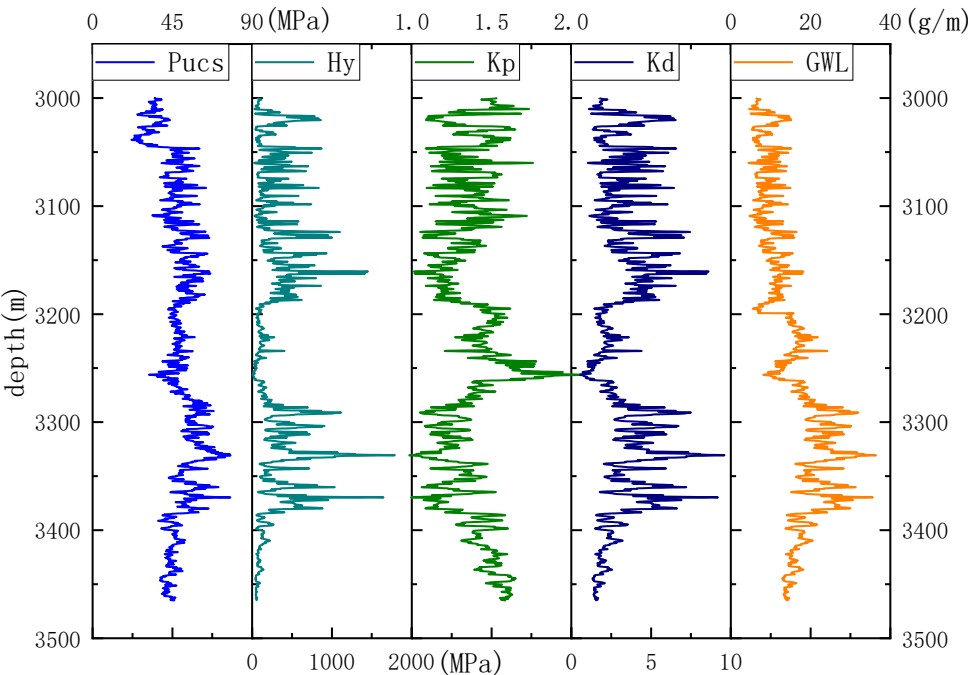

**Figure 9.** Profile of drilling resistance characteristics parameters.

## 4. Drill Bit Selection and Field Application

### 4.1. Drill Bit Failure Analysis and Selection

For a single PDC drill bit, we always hope that both the PDC cutting teeth and the drill bit body have minimal wear at the bottom of the well so that they can maintain normal drilling under complex conditions, ensuring an adequate lifespan for the PDC drill bit. In drilling engineering, mechanical drilling speed, the integrity of PDC cutting teeth, and the drill bit body are typically used as criteria to assess the performance of the drill bit.

An analysis of drill bit failures was conducted based on the usage of drill bits in multiple wells in the Jiyang Depression block during the early stages. Upon on-site inspection of PDC drill bits after they were removed from the well, it was observed that there was severe wear on the outer cone of the drill bit crown, and the PDC teeth exhibited chipping and chunking. Additionally, the drill bit experienced severe diameter reduction, as shown in Figure 10.

Based on the analysis combined with the profile of drillability parameters, the following conclusions can be drawn:

From the condition of the PDC drill bits upon removal from the well, it is evident that there was severe chipping of the PDC teeth in this layer. This is attributed to the strong anisotropy of the strata in this interval and the high hardness of the rocks. The drill bit experienced significant lateral vibration, especially when encountering interbedded formations of varying hardness. The uneven lateral forces exerted on the drill bit led to frequent chipping of the PDC teeth, as they are particularly sensitive to impact.

Measurements revealed that the external diameter of the PDC drill bit wore down by 3 mm. This is due to the strong abrasiveness of the rock formations in this interval. When

the PDC drill bit encounters highly abrasive formations, the external diameter undergoes continuous friction against the surrounding rock due to the rotational and lateral vibrations of the drill bit. Additionally, the high bottom hole pressure and temperature exacerbate the wear of the PDC teeth due to the increased friction with highly abrasive sandstone formations.

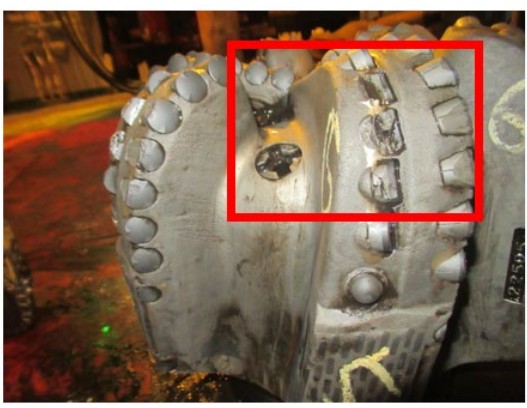 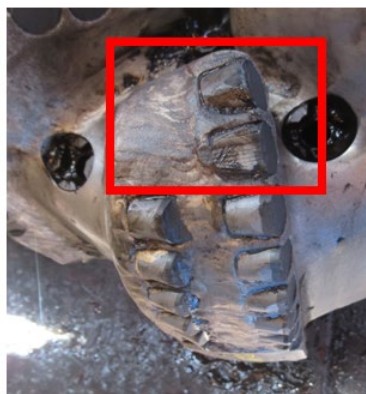

**Figure 10.** Field failure mode of PDC drill bits.

Upon reviewing the drilling records of this drill bit, it was found that the mechanical drilling speed was only 5.8 m/h. This is attributed to the high drillability grade value of the rocks in this interval, making it difficult for the PDC teeth to penetrate into the formation.

After collecting the cuttings from the wellbore and observing them under a microscope, it was found that the main rock types in this interval are sandstone and conglomerate, as shown in Figure 11. It shows the magnified image of the cuttings enlarged 20 times. This lithostratigraphic interval is located at a depth of 3345 m. Upon reviewing the drilling records, it was noted that the drilling rate was relatively low at this interval. This can be attributed to the strong abrasiveness, high hardness, and low plasticity of sandstone and conglomerate rocks, which result in poor drillability and severe wear on the drill bit. The mechanical drilling rate is consequently low. This observation is consistent with the results obtained from the analysis of the drilling resistance characteristics profile. Therefore, it is reasonable to believe that the PDC drill bit used in this interval needs to be further optimized to improve the mechanical drilling rate.

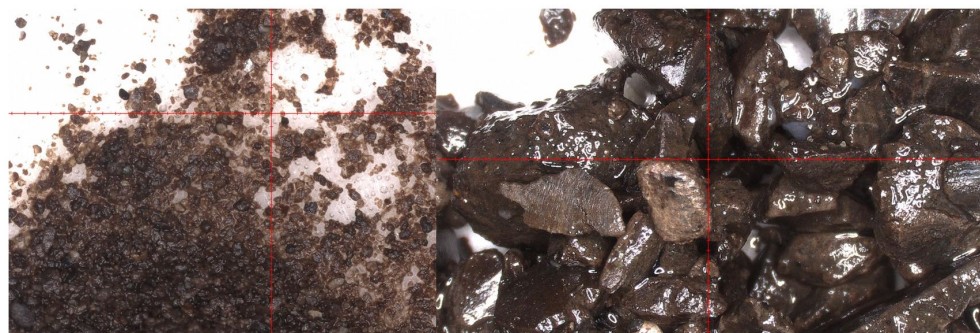

**Figure 11.** Photograph of rock cuttings (enlarged 20 times).

*4.2. Field Application*

In order to reduce the risk of PDC tooth failure and improve the stability of the drill bit during operation, it is necessary to optimize the selection of PDC drill bits to increase mechanical drilling speed and reduce drilling cycle time. Through the analysis of rock drillability parameters and the examination of failed drill bits, it was found that there is significant anisotropy in the rocks within this interval, which generally conform to the characteristics of soft-hard interbedded formations. Therefore, when selecting PDC drill bits, it is recommended to choose small-diameter PDC bits with strong aggressiveness.

Additionally, increasing the density of cutting elements at the crown position and adding more gauge protection elements while enhancing hydraulic action to facilitate cuttings evacuation can improve the wear resistance and impact resistance of PDC teeth. After careful selection, the 12-1/4″ drill bit selected from the drill bit inventory performed well on-site. The appearance of the drill bit after removal from the well is shown in Figure 12. It can be observed from the figure that the overall wear of the drill bit is minimal, with only a 1mm reduction in diameter measured after removal from the well. The drill bit only experienced slight wear at the nose tip. The mechanical drilling speed reached 12.58 m/h, and all selected drill bits successfully penetrated the target interval, achieving the desired drilling effect.

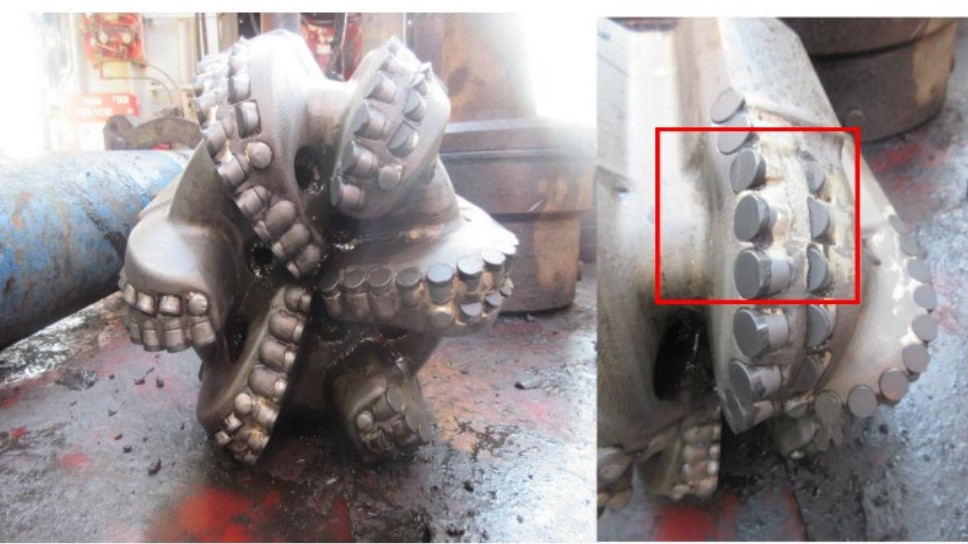

**Figure 12.** Appearance of the drill bit after optimization.

### 5. Conclusions

(1) Based on logging data and laboratory experiments, a profile of rock drilling resistance parameters was established. The results indicate that within the Ji'yang sag formation, the uniaxial compressive strength ranges between 40 and 65 MPa, the hardness ranges between 120 and 1400 MPa, the plasticity coefficient ranges from 1 to 2, the PDC drillability index ranges between 3 and 8, and the abrasiveness index ranges between 15 and 36 g/m. The rocks within this interval are classified as a soft to medium-hard level, with a medium to medium-high level of abrasiveness, and overall exhibit low plasticity based on the plasticity coefficient.

(2) The predominant failure mode of PDC drill bits within this interval is the chipping of PDC teeth at the crown and the wear of gauge protection elements. This is attributed to the strong anisotropy within this interval, causing intense lateral vibrations of the drill bit. The frequent and uneven loading leads to chipping of the PDC teeth. Moreover, the elevated bottom-hole pressure and temperature, coupled with the abrasive nature of the rocks, exacerbate the wear of PDC teeth. Through the observation of rock cuttings after drilling, it was found that at the depth of 3350 m where the mechanical drilling speed of the drill bit was low, the rocks mainly consisted of sandstone and conglomerate. These types of rocks exhibit strong abrasiveness, low plasticity, high compressive strength, and poor drillability, which is consistent with the results obtained through the laboratory experiments. This confirms the credibility of the rock drilling resistance characteristics parameter profile calculated by combining laboratory experiments and logging data, providing guidance for the optimization of drill bits at this depth.

(3) Based on the rock drilling resistance characteristics and the analysis of failed drill bits from previous field operations, selection criteria for PDC drill bits specific to

this formation were proposed. Before optimization, the mechanical drilling speed of the drill bit was only 5.8 m/h. After optimization, the overall wear of the drill bit was minimal, with only slight wear at the crown nose, and the mechanical drilling speed reached 12.58 m/h. The performance of the selected PDC drill bits on-site was satisfactory, resulting in a noticeable increase in drilling speed.

**Author Contributions:** Software, W.C.; Formal analysis, L.Z.; Data curation, X.M. All authors have read and agreed to the published version of the manuscript.

**Funding:** This research received no external funding.

**Data Availability Statement:** Data are contained within the article.

**Conflicts of Interest:** Author Xiaoyong Ma was employed by the company Yellow River Drilling Company, Sinopec Shengli Petroleum Engineering Corporation. The remaining authors declare that the research was conducted in the absence of any commercial or financial relationships that could be construed as a potential conflict of interest. The [company Yellow River Drilling Company, Sinopec Shengli Petroleum Engineering Corporation in affiliation] had no role in the design of the study; in the collection, analyses, or interpretation of data; in the writing of the manuscript, or in the decision to publish the results.

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
