# Peer review of "Study and Application of Rock Drilling Resistance Characteristics in the Jiyang Depression Formation"

_processes, doi:10.3390/pr12050952_

Round 1

Reviewer 1 Report

Comments and Suggestions for Authors

The article contains very interesting laboratory tests relating to the characteristics of PDC drill and their selection based on industrial tests. The presented relationships between acoustic time difference and rock hardness, plasticity rock drillability index and abrasiveness index are important from the point of view of selecting the appropriate drilling tool for specific geological conditions. Below are some comments and suggestions:

1. In the introduction, it should be added some information about the basic construction materials used in drill bits - this is quite important information that determines the durability and choice of the drilling tool;

2. In the subsection 2.1, write down the speed at which the samples were loaded;

3. In the subsection 2.2, please add a literature reference (website to the manufacturer's website) of the equipment used to perform the rock hardness test - so that the characteristics of the control and measurement equipment can be checked;

4. For tables 1 and 2, please add a reference to the literature;

5. In the subsection 2.3, please slightly expand the information about the "i" parameter - Drilling pressure level; first of all, there is no "i" parameter in formula 3;

6. Below table 4, please add explanations for the parameters: Pucs, Hy, GWL and AC (despite the fact that these explanations appear in the text below in the formulas) - it is worth adding them for better readability of the table;

7. For Figure 9, add units on the horizontal axis for the appropriate parameters (both on the lower and upper horizontal axis);

8. In the subsection 3.2, please write whether the value of horizontal stresses affects the drilling results;

9. In the subsection 4.1, based on the literature (several items), it should be indicated what criteria and research methods are adopted to determine the life/wear and selection of the drilling tool - so that the results constitute a form of discussion;

10. In the conclusions, please add one statement regarding the scope of drilling speed for PDC drill bits.

Reviewer 2 Report

Comments and Suggestions for Authors

Minor revision is suggested at this stage.

1. Considering that micro-fractures and the anisotropic nature of rocks can significantly affect drilling operations, a detailed study could help in understanding their precise impacts and in improving bit selection and drilling strategies.

2. Consider consulting more chemical-related processes and discuss, the article below is suggested to be consulted as a starting point. Status and outlook of oil field chemistry-assisted analysis during the energy transition period. Energy & Fuels. 2022.

3. Since clay content can influence the mechanical properties of rocks, quantifying this relationship could provide more accurate predictive models for drilling operations.

4. Research into customized bit designs considering specific geological characteristics could lead to significant improvements in drilling efficiency and cost-effectiveness.

5.  A study focusing on wear and tear and the long-term economic impacts of drilling in such conditions could be crucial for operational planning and budgeting.

Comments on the Quality of English Language

English fine
